# Upregulation of 15-Hydroxyprostaglandin Dehydrogenase by Celecoxib to Reduce Pain After Laparoendoscopic Single-Site Surgery (POPCORN Trial): A Randomized Controlled Trial

**DOI:** 10.3390/biomedicines13071784

**Published:** 2025-07-21

**Authors:** Kyung Hee Han, Sunwoo Park, Seungmee Lee, Jiyeon Ham, Whasun Lim, Gwonhwa Song, Hee Seung Kim

**Affiliations:** 1Department of Obstetrics and Gynecology, Seoul National University Hospital, Seoul 03080, Republic of Korea; kyunghhan@gmail.com; 2Department of Plant & Biomaterials Science, Gyeongsang National University, Jinju-si 52725, Republic of Korea; sw.park@gnu.ac.kr; 3Department of Obstetrics and Gynecology, Keimyung University School of Medicine, Daegu 41931, Republic of Korea; seungmeemd@gmail.com; 4Department of Animal Science and Biotechnology, Chungnam National University, Daejeon 34134, Republic of Korea; jyham@cnu.ac.kr; 5Department of Biological Sciences, College of Science, Sungkyunkwan University, Suwon 16419, Republic of Korea; wlim@skku.edu; 6Department of Biotechnology, College of Life Sciences and Biotechnology, Korea University, Seoul 02841, Republic of Korea; 7Institute of Reproductive Medicine and Population, Medical Research Center, Seoul National University, Seoul 03087, Republic of Korea; 8Department of Obstetrics and Gynecology, Seoul National University College of Medicine, Seoul 03080, Republic of Korea

**Keywords:** celecoxib, postoperative pain, carbon dioxide, 5-hydroxyprostaglandin dehydrogenase, laparoendoscopic single-site surgery

## Abstract

**Background**: Peritoneal stretching from CO_2_ insufflation is a primary mechanism of pain associated with laparoscopy. Cyclooxygenase-2 inhibitors are promising anti-inflammatory and analgesic agents. This study aimed to evaluate the effect of celecoxib on postoperative pain reduction and associated changes in peritoneal gene expression after laparoendoscopic single-site (LESS) surgery for benign gynecologic disease. **Methods**: In this randomized, double-blind, placebo-controlled pilot study, 70 patients were randomly assigned to receive either celecoxib or placebo (400 mg) 40 min before surgery. Peritoneal tissues were collected before and after CO_2_ insufflation. We analyzed changes in expressions of prostaglandin I_2_ synthase, prostaglandin E synthase (*PTGES*), *PTGES3*, aldo-keto reductase family 1 member C1, and 15-hydroxyprostaglandin dehydrogenase (*HPGD*). Numeric Rating Scale (NRS) pain scores were also compared between groups. **Results**: A total of 62 patients completed the study: 30 in the celecoxib group and 32 in the placebo group. The mean CO_2_ exposure time was 60.4 min. In a quantitative real-time polymerase chain reaction analysis, *HPGD* mRNA expression significantly increased after surgery in patients exposed to CO_2_ for more than 60 min. Patients treated with celecoxib showed a significantly higher rate of grade 3 expression (83.3% vs. 37.5%; *p* = 0.01) and a level 2 increase in *HPGD* expression on in situ hybridization (58.3% vs. 12.5%; *p* = 0.01), despite no significant difference on immunohistochemistry. Moreover, celecoxib effectively reduced NRS pain scores compared to placebo. **Conclusions**: In this pilot study, celecoxib appeared to reduce postoperative pain and was associated with increased HPGD mRNA expression in the peritoneal tissue of patients with prolonged CO_2_ exposure during LESS surgery. These exploratory findings warrant confirmation in larger trials with functional validation of HPGD expression (ClinicalTrials.gov, NCT03391570).

## 1. Introduction

Postoperative pain management remains a significant challenge in laparoscopic surgery, with approximately 80% of patients experiencing severe pain that requires intervention despite the procedure’s advantages, such as reduced analgesic use, improved cosmetic outcome, and faster recovery [1,2,3]. While patient-controlled analgesia (PCA), including opioids, is commonly employed to manage acute surgical pain, its efficacy is often limited by frequent postoperative nausea and vomiting (PONV) [4,5].

The etiology of postoperative pain following laparoscopic surgery is multifaceted and not yet fully understood. However, several factors may contribute to pain sensation, including physical stimulation by surgical incision [6], tissue trauma caused by peritoneal stretching during carbon dioxide (CO_2_) insufflation [7], and subsequent inflammation leading to peripheral nociceptor stimulation [8]. Among the various approaches to pain management after laparoscopic procedures, cyclooxygenase-2 (COX-2) inhibitors have emerged as promising anti-inflammatory and analgesic agents. These are believed to effectively reduce pain by inhibiting prostanoid production [9,10].

Generally, COX-2 synthesizes prostanoids from arachidonic acid, released from membrane phospholipids in response to inflammation. This process may increase the electrical excitability of sensory neurons by modulating neurotransmitter release [11]. Therefore, COX-2 inhibitors are considered effective in reducing postoperative pain after laparoscopic surgery by inhibiting prostanoid synthesis. In addition, COX-2 inhibitors are anticipated to regress preneoplastic lesions and suppress disease progression [12].

However, the precise mechanisms by which COX-2 inhibitors effectively alleviate pain after laparoscopic surgery remain unclear, particularly considering the diverse causes of pain and the complex signaling pathways involved in COX-2 inhibition. To address this, we conducted a randomized, double-blind, placebo-controlled pilot trial, to investigate the Pain reduction by celecOxib via increased 15-hydroxyProstaglandin dehydrogenase expression after laparoendosCOpic single-site (LESS) suRgery for benign gyNecologic diseases (POPCORN). The primary aim was to investigate changes in gene expression related to prostanoid metabolism in peritoneal tissue, modulated by celecoxib, a COX-2 inhibitor, and their role in reducing postoperative pain following LESS surgery for benign gynecologic diseases.

## 2. Materials and Methods

### 2.1. Study Design

This study was designed as a randomized, double-blind, placebo-controlled pilot study. Prior to its commencement, the study protocol received approval from the Institutional Review Board (No. 1705-061-853). Moreover, this trial was registered on ClinicalTrial.gov (No. NCT03391570). An independent data and safety monitoring committee oversaw the study’s progress and ensured adherence to safety protocols throughout its duration.

### 2.2. Study Population

Patient recruitment occurred between February and November 2018 in Seoul National University Hospital, based on the following eligibility criteria: age 20 years or older; diagnosed with benign gynecologic conditions suitable for LESS surgery; American Society of Anesthesiologists (ASA) physical status classification of 1 or 2; and provision of written informed consent. Moreover, the following criteria were excluded: suspected malignancy on imaging studies; peritoneal inflammatory conditions such as endometriosis and rheumatic diseases; history of severe adhesion or peritonitis warranting multi-port laparoscopic or open surgery; contraindications to COX-2 inhibitors.

### 2.3. Sample Size

In the absence of existing data on COX-2 inhibitors’ impact on gene expression during LESS procedures, we established preliminary assumptions regarding gene expression rate. Specifically, we anticipated a 50% high expression rate in the celecoxib group and a 15% low expression rate in the placebo group. To ensure robust statistical validity, we designed our study with a superiority test featuring 80% statistical power and a two-sided significance level of 5%. Additionally, we incorporated a 10% potential non-compliance buffer into our sample size calculation. Based on these assumptions, a minimum of 62 patients was required to detect significant differences in gene expression between groups.

### 2.4. Randomization

Patients were randomly assigned to either the celecoxib or placebo group in a 1:1 ratio. An independent investigator (SL) managed the randomization process using a web-based program to generate a simple randomization table without blocking. To maintain blinding, the investigator ensured that the physicians were unaware of the randomization sequence and concealed the table with an opaque covering to prevent anticipation of subsequent assignments. Patients remained blind to their group allocation throughout the duration of the trial until its completion.

### 2.5. Surgical Procedures

In this study, we used 400 mg of celecoxib to evaluate the role of a COX-2 inhibitor on postoperative pain because some studies had shown that 400 mg of celecoxib might be more effective than its baseline dose (200 mg) [13,14]. Following randomization, patients received either 400 mg of celecoxib or a matching placebo 40 min before surgery. LESS was performed using DreamOneport^®^ (Dreampac Corporation, Wonju, Republic of Korea) inserted through the umbilicus, with capnoperitoneum maintained at 12 mmHg via CO_2_ insufflation throughout the procedure. We collected two 2 × 2 cm peritoneal tissue samples at two time points: before the main surgery to capture gene expressions related to prostanoid metabolism prior to CO_2_ exposure, and after completing LESS to reflect gene expression changes induced by CO_2_ exposure. Each set of samples was divided, with half immediately frozen in liquid nitrogen for mRNA expression analysis via quantitative reverse transcription-polymerase chain reaction (qRT-PCR), and the other half prepared as paraffin-embedded tissue blocks for protein and mRNA expression studies using immunohistochemistry (IHC) and in situ hybridization (ISH), respectively.

During the induction of general anesthesia, an initial intravenous dose of fentanyl ranging from 50 to 100 micrograms was administered. Throughout the surgery, additional doses of fentanyl were given at intervals of 30 to 60 min, depending on the patient’s needs and the duration of the procedure. After the surgery, once the patients regained consciousness, PCA was implemented to manage postoperative pain in all patients. For PCA administration, 100 micrograms of fentanyl was diluted in 2 mL of normal saline and administered intravenously. Additionally, a continuous infusion was prepared by diluting 1000 micrograms of fentanyl in 80 mL of normal saline, which was delivered at a constant rate. To allow the patient to manage breakthrough pain, the PCA device was programmed so that the patient could self-administer a 50-microgram bolus dose of fentanyl when needed. This approach provided both continuous baseline pain control and the flexibility for the patient to address acute pain episodes effectively. Moreover, PICA was removed if PONV occurred.

### 2.6. Quantitative Real-Time Polymerase Chain Reaction

mRNA for qRT-PCR was extracted from peritoneal tissue samples obtained during LESS. Gene expression levels were quantified using SYBR Green (Sigma-Aldrich, St. Louis, MO, USA) on an Applied Biosystems Real-Time PCR System, with ROX dye serving as a negative control. The PCR protocol consisted of an initial denaturation of 95 °C for 3 min, followed by 40 cycles at 95 °C for 30 s, 60 °C for 30 s, and 72 °C for 30 s. The relative gene expression was calculated using the 2^–ΔΔCT^ method, with CT values normalized to GAPDH mRNA expression as an internal control. To ensure primer specificity, melt curve analysis was performed for each target gene.

### 2.7. Immunohistochemistry

IHC was used to visualize the expression of relevant proteins in peritoneal tissue samples. Antigen retrieval was performed on slides prepared from paraffin-embedded peritoneal tissue blocks using the citrate boiling method. The slides were then incubated overnight at 4 °C with 1 mL of mouse monoclonal primary antibodies at a 1:500 dilution. For negative controls, slides were incubated with rabbit immunoglobulin G at the same concentration. All slides were then treated with 1 mL of biotinylated secondary antibody for one hour at room temperature, washed with phosphate-buffered saline, dehydrated with ethanol, and cleared with xylene. Images were captured using a Leica DM 2500 microscope (Leica Microsystems GmbH, Wetzlar, Germany).

### 2.8. In Situ Hybridization

ISH was employed to visualize the expression of relevant mRNAs in peritoneal tissue samples. The process began with the amplification of specific partial cDNA sequences for the genes of interest using qRT-PCR. These amplified sequences were then extracted from agarose gel and inserted into the TOPO TA cloning vector (Invitrogen, Thermo Fisher Scientific, Waltham, MA, USA).

Slides prepared from paraffin-embedded tissue blocks were deparaffinized and subjected to a pre-hybridization step for 10 min using a solution of 50% formamide and 4X standard saline citrate. RNA probes labeled with digoxigenin were transcribed using the DIG RNA labeling kit (Roche Diagnostics, Basel, Switzerland). The probes were then applied to the tissue slides and incubated for 16 h.

Following hybridization, the signal was detected using sheep anti-DIG antibodies. The chromogenic reaction was developed using a mixture containing 0.4 mM 5-bromo-4-chloro-3-indolyl phosphate, 0.4 mM nitroblue tetrazolium, and 2 mM levamisole. The stained tissue sections were subsequently examined using a Leica DM 2500 microscope.

### 2.9. Endpoints

The primary endpoints of this study were changes in gene expression related to prostanoid metabolism, influenced by time-dependent CO_2_ exposure during LESS. We compared the mRNA expression of candidate genes involved in prostanoid metabolism in the peritoneal tissue before and after CO_2_ exposure using qRT-PCR. Subsequently, we selected the genes expressed differently according to time-dependent CO_2_ exposure in the peritoneal tissue by IHC and ISH. Secondary endpoints included numeric rating scale (NRS) pain scores (range: 0–10) and various postoperative outcomes such as operation time, estimated blood loss (EBL), transfusion requirements, frequency of rescue analgesic use, discontinuation of PCA, and length of hospitalization between the celecoxib and placebo groups. Demographic data, surgical history, procedural details, and postoperative outcomes were collected. NRS pain scores were evaluated at regular intervals (0, 6, 12, 18, 24, 30, 36, 42, and 48 h) following LESS.

### 2.10. Statistical Analysis

We performed statistical analyses on both the intention-to-treat (ITT) and per-protocol (PP) populations. For categorical variables, we employed Fisher’s exact test or chi-square test. Continuous variables were analyzed using either the Mann-Whitney U test or Student’s T-test, based on data distribution. NRS pain scores across time points were compared using repeated measures analysis of variance (ANOVA). All statistical analyses were conducted using SPSS version 24.0 (SPSS, Inc., Chicago, IL, USA) with statistical significance set at *p* < 0.05.

## 3. Results

### 3.1. Clinicopathologic Characteristics

A total of 70 patients were initially enrolled in this study. However, eight patients were subsequently excluded: six due to withdrawal of written consent, and two due to disagreement with postoperative pain assessment protocols. This resulted in 62 patients being randomly assigned for ITT analysis. Of these, 52 patients’ data were used for PP analysis, as we were unable to assess NRS pain scores at certain postoperative time points for 10 (Figure 1).

The study population had a mean age of 49.6 years, a mean BMI of 22.8 kg/m^2^, and an average peritoneal tissue exposure time to CO_2_ of 60.4 min. When comparing baseline characteristics between the celecoxib group (n = 30) and the placebo group (n = 32), no significant differences were observed (Table 1).

### 3.2. Quantitative Real-Time Polymerase Chain Reaction

While previous studies have reported significant expression of genes related to prostanoid metabolism in malignant or inflammatory tissues such as endometriosis and rheumatic diseases [15,16,17], we found no relevant research evaluating these gene expressions in normal peritoneal tissues in the context of postoperative pain after LESS. To identify candidate genes in the prostanoid biosynthesis pathway potentially related to pain after LESS, we conducted a preliminary investigation prior to this trial. Fresh peritoneal tissue samples were obtained from ten patients undergoing LESS hysterectomy for uterine fibroids. Samples were collected before and after CO_2_ exposure. Of these patients, five received 400 mg of celexocib preoperatively, while the other five received a placebo.

To identify candidate genes potentially affected by CO_2_ exposure during LESS, we screened 12 genes related to prostanoid metabolism in paired peritoneal tissue samples collected from five patients. Using qRT-PCR, we examined the expression of phospholipase A2 group IVA (*PLA2G4A*), prostaglandin-endoperoxide synthase 1 (*PTGS1*), PTGS2, prostaglandin E synthase (*PTGES*), *PTGES2*, *PTGES3*, prostaglandin I2 synthase (*PTGIS*), aldo-keto reductase family 1 member C1 (*AKR1C1*), *AKR1C2*, *AKR1C3*, thromboxane A synthase (*TBAS*), 15-hydroxyprostaglandin dehydrogenase (*HPGD*). Our results showed that *PLA2G4A*, *PTGS2*, *AKR1C1*, *AKR1C2*, and *TBAS* were not detected in the samples, while *PTGS1* and *PTGES2* were excluded due to the formation of the dimer. Based on these preliminary findings, we selected *PTGIS*, *PTGES*, *PTGES3*, *HPGD*, and *AKR1C3* as candidate genes for further investigation in our main study, as they demonstrated potential expression changes in response to CO_2_ exposure during LESS (Figure 2).

We subsequently evaluated changes in the expression of the five candidate genes in 62 patients enrolled in this study by using qRT-PCR. The primer sequences for the candidate genes were synthesized by Bioneer (Daejeon, South Korea; Table 2). Our analysis revealed no significant changes in the expression of these five genes before and after CO_2_ exposure in the placebo group (Figure 3). In the celecoxib group, we initially observed no significant changes when analyzing all patients (n = 30). However, in patients exposed to CO_2_ for more than 60 min (n = 12), we detected a significant increase in the mRNA expression of *HPGD* in peritoneal tissues following LESS (Figure 4).

### 3.3. Immunohistochemistry and In Situ Hybridization

We conducted IHC and ISH for *HPGD* on peritoneal tissue samples from 28 patients who underwent LESS surgery with CO_2_ exposure for more than one hour during LESS surgery in both the celecoxib (n = 12) and placebo groups (n = 16). The staining intensity of vascular endothelial cells was graded using a three-tier system: no or weak expression, grade 1; moderate expression, grade 2; strong expression, grade 3. Interestingly, our IHC analysis revealed that all peritoneal tissue samples, regardless of whether they were collected before or after CO_2_ exposure during LESS surgery, showed no detectable expression of HPGD.

For ISH analysis, we amplified specific partial cDNA sequences for *HPGD* (GenBank no. L76465.1) using qRT-PCR and extracted them from agarose gel. These partial cDNAs were then inserted into the TOPO TA cloning vector (Invitrogen, Thermo Fisher Scientific, MA, USA). The plasmids containing *HPGD* were amplified using T7 (5′-TGT AAT ACG ACT CAC TAT AGG G-3′) and SP6 (5′-CTA TTT AGG TGA CAC TAT AGA AT-3′) primers.

Our ISH results revealed no difference in *HPGD* staining intensity in peritoneal tissues before CO_2_ exposure between the celecoxib and placebo groups. However, after CO_2_ exposure, grade 3 expression of *HPGD* was significantly more prevalent in the celecoxib group compared to the placebo group (83.3% vs. 37.5%; *p* = 0.01). To evaluate changes in *HPGD* expression before and after CO_2_ exposure, we defined two levels of HPGD expression increase: a change from grade 1 to grade 2 expression, or from grade 2 to grade 3 expression, level 1 increase; a change from grade 1 to grade 3 expression, level 2 increase. The celecoxib group demonstrated a significantly higher rate of level 2 increase than the placebo group (58.3% vs. 12.5%; *p* = 0.01), suggesting that celecoxib may enhance *HPGD* expression in peritoneal tissues in response to CO_2_ exposure during laparoscopic surgery (Table 3, Figure 5).

### 3.4. Numeric Rating Scale Pain Scores and Postoperative Outcomes

We compared NRS pain scores after LESS between the celecoxib and placebo groups (Table 4). While the ITT population showed no significant differences, the PP population demonstrated lower NRS pain scores in the celecoxib group at 6 and 12 h after surgery (Figure 6).

Subgroup analyses based on CO_2_ exposure time revealed no differences between groups for patients exposed to CO_2_ for less than 60 min in both ITT and PP populations. However, for patients exposed to CO_2_ for more than 60 min, celecoxib was associated with significantly lower NRS scores in both ITT and PP populations (Figure 7, Table 5 and Table 6).

Table 7 compares postoperative outcomes between the celecoxib and placebo groups, which demonstrated no differences in operation time, EBL, number of transfusions and rescue analgesics, discontinuation of PCA use due to PONV, and duration of hospitalization.

## 4. Discussion

In this study, we found that oral administration of 400 mg celecoxib 40 min before LESS surgery enhanced *HPGD* expression in peritoneal tissues with a reduction of postoperative pain, especially when CO_2_ exposure time exceeded one hour. To the best of our knowledge, this is the first evidence that celecoxib may increase *HPGD* expression, which is associated with reduced postoperative pain after laparoscopic surgery.

Our previous research showed no difference in abdominal pain between LESS and vaginal natural orifice transluminal endoscopic surgery, which avoids abdominal incisions, suggesting that CO_2_ exposure may significantly contribute to postoperative pain [8]. However, some argue that multiple incisions cause greater pain than a single incision [18]. Therefore, we focused on LESS to minimize incision-related pain and propose that time-dependent CO_2_ exposure is associated with pain severity.

Considering that pneumoperitoneum induced by CO_2_ insufflation may alter prostanoid metabolism, including immune responses mediated by macrophage function in the peritoneal tissue [19], pro-inflammatory polarization of peritoneal macrophages may induce prostaglandin production [20], and celecoxib might be involved in prostanoid metabolism in normal peritoneal tissues exposed to CO_2_ during laparoscopic surgery. Moreover, we excluded peritoneal inflammatory diseases such as malignancies, endometriosis, and rheumatic diseases because these conditions can elevate gene expressions in the prostanoid biosynthesis pathway. In various malignancies, inflammatory cells present in the tumor microenvironment may contribute to carcinogenesis and tumor proliferation through the overexpression of genes involved in prostanoid metabolism.

Among the five candidate genes, *HPGD* expression increased with celecoxib in CO_2_-exposed normal peritoneal tissues. *HPGD* degrades prostaglandins and is often downregulated in high prostaglandin states associated with severe inflammation and oncogenesis [12,21]. Consequently, celecoxib has been associated with increased *HPGD* expression, suggesting it may serve as a biomarker or therapeutic target for reducing the risk of malignancies and the severity of endometriosis [21,22,23]. Consistent with previous studies, we observed that celecoxib was linked to higher *HPGD* expression in normal peritoneal tissues compared to the placebo group. This finding suggests that postoperative pain may be alleviated by promoting prostaglandin degradation via *HPGD* activation, rather than solely by inhibiting their production through COX inhibition.

However, the increased *HPGD* expression in normal peritoneal tissues may be lower than that observed in malignant or inflammatory tissues, as this study did not demonstrate *HPGD* protein expression by IHC, unlike previous findings in cancer or endometriosis [21,23]. It can be postulated that CO_2_ exposure during laparoscopic surgery may influence the analgesic effects of celecoxib, with a time-dependent threshold determining their onset. In the POPCORN trial, this threshold was identified as exceeding 60 min of CO_2_ exposure, as celecoxib exhibited analgesic effects only beyond this duration. This hypothesis is further supported by previous studies indicating that COX inhibitors were ineffective in reducing postoperative pain following short-duration laparoscopic procedures [24,25].

Limitations of our study are as follows. First, the POPCORN trial was a pilot study due to the lack of sample size determination. Although the level 2 increase in *HPGD* expression aligned with expectations between the celecoxib and placebo groups, this comparison was based on only 28 patients exposed to CO_2_ for more than 60 min, which is less than half of the calculated sample size. Since there was no validation by repeated assessments, the results of the POPCONRN trial are limited by low statistical power and the possibility of false positives. Second, the results of this study may be affected by variations in the average duration of CO_2_ exposure due to variations in the number of patients. Thus, it is necessary to validate the results in a clinical trial with a larger number of patients. Third, the initial screening to identify candidate genes related to CO_2_ exposure was underpowered due to a limited number of patients in this pilot study, and we could not conclude that some genes, including PTGES, PTGIS, and AKR1C3, are not necessarily induced by CO_2_ exposure. Fourth, we can only confirm that *HPGD* gene expression may be upregulated with increasing CO_2_ exposure time in the peritoneum, but we cannot conclude whether this results in direct activation of the *HPGD* gene via increased CO_2_ exposure time because we did not conduct the measurement of post-transcriptional regulation of HPGD by increased mRNA expression.

This study also has some strengths. First, it provides initial evidence that celecoxib, a COX-2 inhibitor, may exert analgesic effects by enhancing *HPGD* expression, which is associated with prostaglandin degradation. Second, the study exclusively employed LESS, minimizing the potential bias of pain severity evaluation that could arise from time-dependent CO_2_ exposure associated with multiple surgical incisions. Third, we excluded other factors that could influence gene expression related to prostanoid metabolism, such as inflammatory or malignant diseases. Last, the trial utilized a double-blind, randomized controlled design to strengthen clinical evidence regarding postoperative pain assessment.

## 5. Conclusions

This Celecoxib may upregulate *HPGD*, promoting prostaglandin elimination and thereby reducing postoperative pain in patients undergoing LESS for benign gynecologic conditions. Furthermore, the analgesic effect of celecoxib appears particularly notable with CO_2_ exposure exceeding one hour.

## Figures and Tables

**Figure 1 biomedicines-13-01784-f001:**
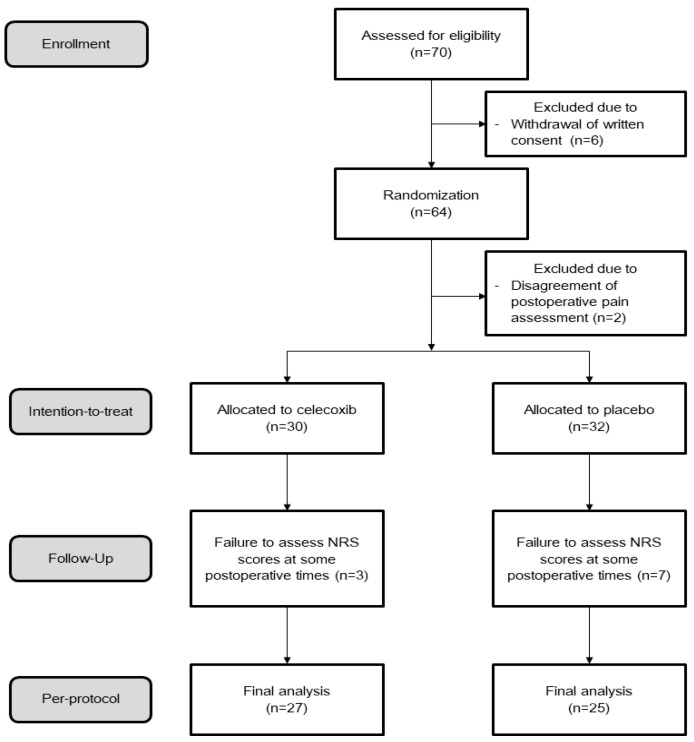
Flow of patient enrollment; NRS indicates Numeric Rating Scale.

**Figure 2 biomedicines-13-01784-f002:**
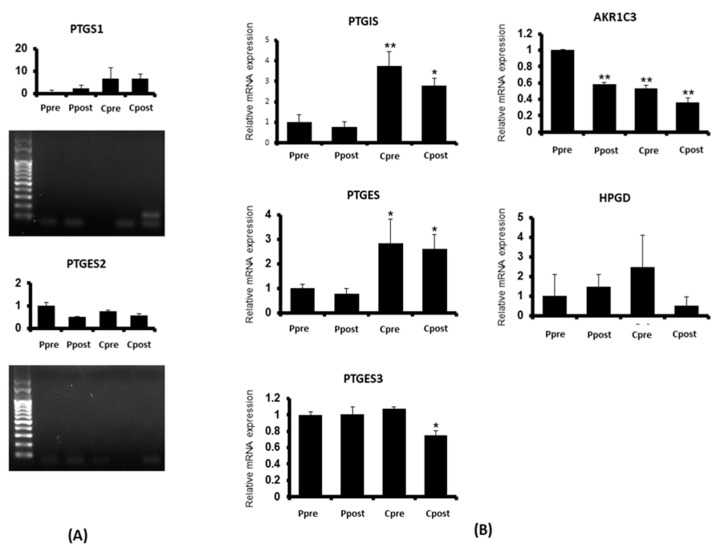
Changes in gene expression induced by CO_2_ exposure during laparoendoscopic single-site surgery (**A**) Prostaglandin-endoperoxide synthase 1 (*PTGS1*) and prostaglandin E synthase 2 (*PTGES2*) genes were excluded due to the formation of dimer; (**B**) prostaglandin I2 synthase (*PTGIS*), *PTGES*, *PTGES3*, aldo-keto reductase family 1 member C3 (*AKR1C3*) and 15-hydroxyprostaglandin dehydrogenase (HPGD) were selected as candidate genes which expressions could be changed by exposure to CO_2_ during laparoendoscopic single-site surgery (Ppre, preoperative after taking placebo 400 mg; Ppost, postoperative after taking placebo 400 mg; Cpre, preoperative after taking celecoxib 400 mg; Cpost, postoperative after taking celecoxib 400 mg; * *p* < 0.05; ** *p* < 0.01).

**Figure 3 biomedicines-13-01784-f003:**
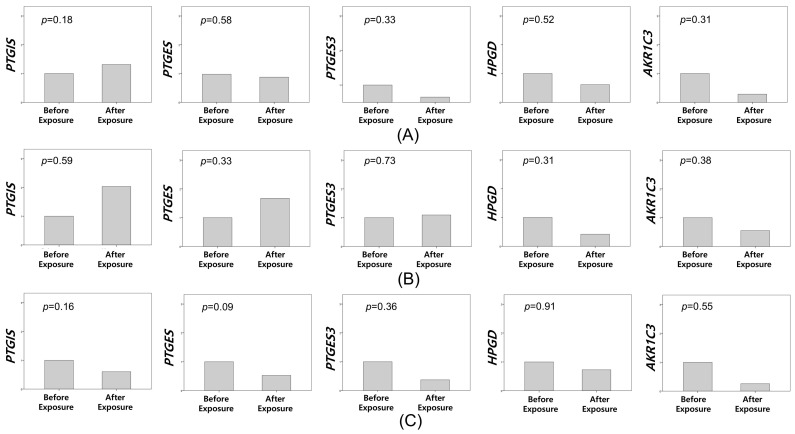
Changes in the Expression of Five Genes Before and After CO_2_ Exposure in the Placebo Group. Changes in prostaglandin I2 synthase (*PTGIS*), prostaglandin E synthase (*PTGES*), *PTGES3*, aldo-keto reductase family 1 member C3 (*AKR1C3*), and 15-hydroxyprostaglandin dehydrogenase (*HPGD*) expressions before and after exposure to CO_2_ in the placebo group; (**A**) all patients (n = 32); (**B**) those exposed to CO_2_ for less than one hour (n = 16); (**C**) those exposed to CO_2_ for more than one hour (n = 16).

**Figure 4 biomedicines-13-01784-f004:**
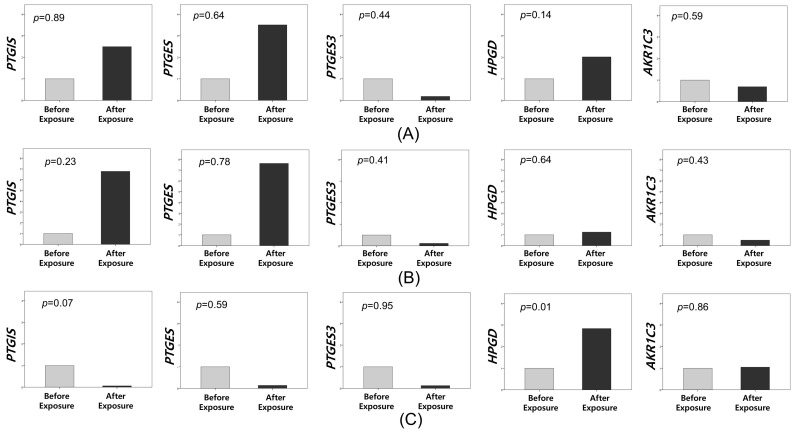
Changes in prostaglandin I2 synthase (*PTGIS*), prostaglandin E synthase (*PTGES*), *PTGES3*, aldo-keto reductase family 1 member C3 (*AKR1C3*), and 15-hydroxyprostaglandin dehydrogenase (*HPGD*) expressions before and after exposure to CO_2_ in the celecoxib group; (**A**) all patients (n = 30); (**B**) those exposed to CO_2_ for less than one hour (n = 18); (**C**) those exposed to CO_2_ for more than one hour (n = 12).

**Figure 5 biomedicines-13-01784-f005:**
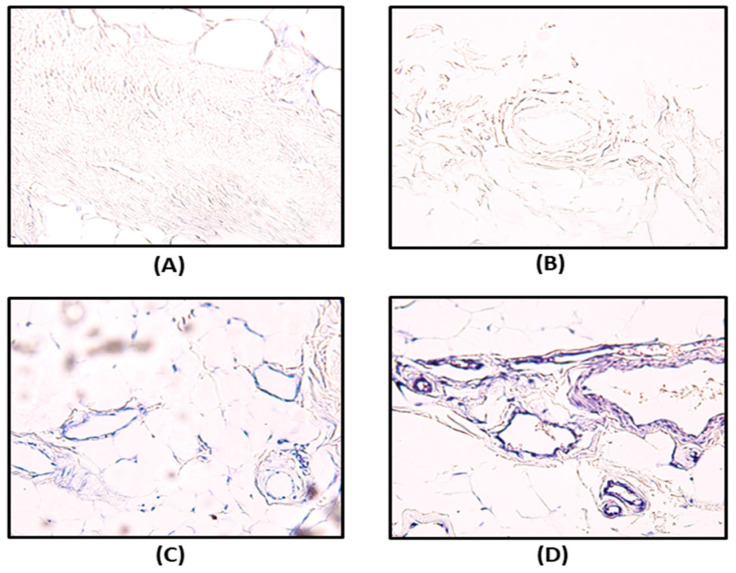
Immunohistochemistry (IHC) and in situ hybridization (ISH) for 15-hydroxyprostaglandin dehydrogenase (*HPGD*) on peritoneal tissues of only 28 patients exposed to CO_2_ for more than one hour in the celecoxib and placebo groups: (**A**) no or weak expression (grade 1) on IHC; (**B**) no or weak expression (grade 1) on ISH; (**C**) moderate expression (grade 2) on ISH; (**D**) strong expression (grade 3) on ISH.

**Figure 6 biomedicines-13-01784-f006:**
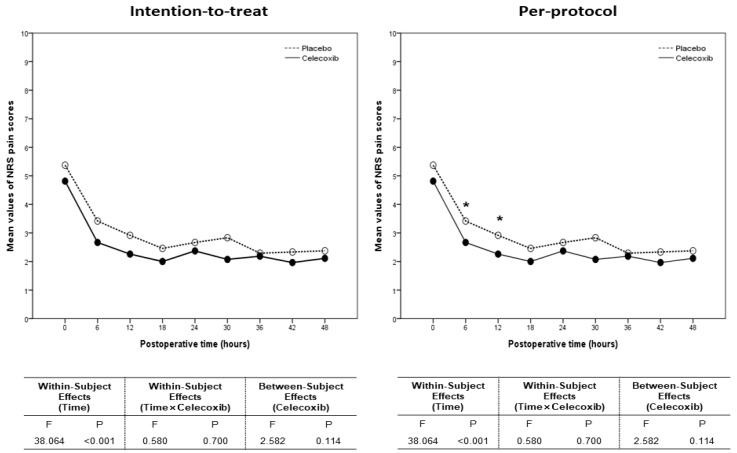
Comparison of NRS pain scores after laparoendoscopic single-site surgery between the celecoxib and placebo groups in the intention-to-treat and per-protocol populations (* *p* < 0.05).

**Figure 7 biomedicines-13-01784-f007:**
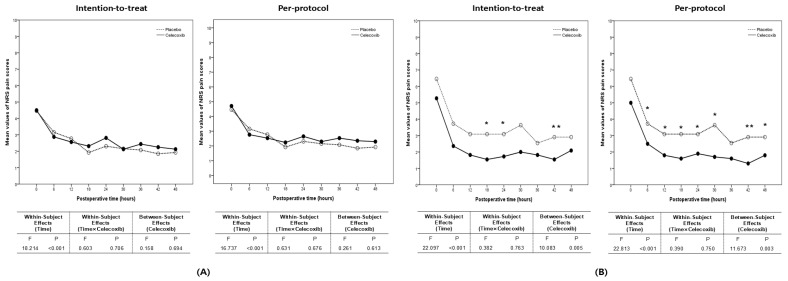
Comparison of NRS pain scores after laparoendoscopic single-site surgery between the celecoxib and placebo groups in the intention-to-treat and per-protocol populations: (**A**) exposure to CO_2_ for less than one hour; (**B**) exposure to CO_2_ for more than one hour (* *p* < 0.05; ** *p* < 0.01).

**Table 1 biomedicines-13-01784-t001:** Baseline characteristics.

Characteristics	Celecoxib (n = 30)	Placebo (n = 32)	*p* Value
Age (year, mean, SD)	49.1 ± 9.6	50 ± 11.6	0.73
BMI (kg/m^2^, mean, SD)	23 ± 2.9	22.7 ± 3.1	0.69
Previous surgery (n, %)	17 (53.1)	11 (36.7)	0.21
Types of surgical procedures (n, %)			0.80
Unilateral salpingo-oophorectomy	3 (10)	1 (3.1)	
Bilateral salpingo-oophorectomy	5 (16.7)	5 (15.6)	
Unilateral ovarian cystectomy	1 (3.3)	3 (9.4)	
Bilateral ovarian cystectomy	1 (3.3)	1 (3.1)	
Hysterectomy	16 (53.3)	16 (50)	
Myomectomy	4 (13.3)	6 (18.8)	
The exposure time of peritoneal tissues to CO_2_ (minutes, mean, SD)	60.3 ± 54.7	60.5 ± 50.5	0.59

Abbreviation: BMI, body mass index; CO_2_, carbon dioxide; SD, standard deviation.

**Table 2 biomedicines-13-01784-t002:** Primer sequence information for quantitative real-time polymerase chain reaction.

Gene	GenBank No.	Forward	Reverse
*PTGIS*	BC101811.1	CATGTGCAGTGTCAAAAGTCG	TGCATCTCCTCTGACACACC
*PTGES*	BC008280.1	GGTCTTGGGTTCCTGTATGG	AAAGACATCCAAAGCCAACG
*PTGES3*	BC003005.1	TGGGGCAATTTTAAGTCAGC	TTGCCTAGGACCTCAACAGC
*HPGD*	L76465.1	GGCATAGTTGGATTCACACG	AACAAAGCCTGGACAAATGG
*AKR* *1* *C3*	BT007286.1	AGTACAAGCCTGTCTGCAACC	TCTCGTTGAGATCCCAGAGC

Abbreviations: *PTGIS*, prostaglandin I2 synthase; *PTGES*, prostaglandin E synthase; *PTGES3*, prostaglandin E synthase 3; *AKR1C3*, aldo-keto reductase family 1 member C3; *HPGD*, 15-hydroxyprostaglandin dehydrogenase.

**Table 3 biomedicines-13-01784-t003:** Expression of 15-hydroxyprostaglandin dehydrogenase (*HPGD*) and its changes on in situ hybridization in 28 patients exposed to CO_2_ for more than one hour who took celecoxib or placebo before surgery.

**(A) *HPGD* Expression**						
**Grade**	**Before Exposure to CO_2_**		***p* Value**	**After Exposure to CO_2_**		***p* Value**
	**Celecoxib** **(n = 12, %)**	**Placebo** **(n = 16, %)**		**Celecoxib** **(n = 12, %)**	**Placebo** **(n = 16, %)**	
Grade 1 (no or weak)	9 (75)	10 (62.5)	0.78	0 (0)	0 (0)	0.01
Grade 2 (moderate)	2 (16.7)	4 (25)		2 (16.7)	10 (62.5)	
Grade 3 (strong)	1 (8.3)	2 (12.5)		10 (83.3)	6 (37.5)	
**(B) Changes in *HPGD* Expression Before and After CO_2_ Exposure**						
**Degree of Change**	**Celecoxib** **(n = 12, %)**		**Placebo** **(n = 16, %)**			***p* Value**
Level 1 increase *	5 (41.7)		14 (87.5)			0.01
Level 2 Increase ^†^	7 (58.3)		2 (12.5)			

* Level 1 increase was defined as a change from grade 1 to grade 2 expression, or from grade 2 to grade 3 expression. ^†^ Level 2 increase was defined as a change from grade 1 to grade 3 expression.

**Table 4 biomedicines-13-01784-t004:** Comparison of NRS pain scores after laparoendoscopic single-site surgery in all patients.

Postoperative Time	Intention-to-Treat Population			Per Protocol Population		
	Celecoxib (n = 30)	Placebo (n = 32)	*p* Value	Celecoxib (n = 27)	Placebo (n = 25)	*p* Value
0 h	4.5 (1, 9)	5 (3, 10)	0.51	4 (1, 9)	5 (3, 10)	0.35
6 h	2.5 (0, 5)	3 (0, 5)	0.18	2 (1, 5)	3 (1, 5)	0.03
12 h	2 (1, 6)	3 (1, 5)	0.10	2 (1, 6)	3 (1, 5)	0.03
18 h	2 (1, 5)	2 (0, 6)	0.23	2 (1, 4)	2 (0, 6)	0.27
24 h	2 (0, 7)	2.5 (0, 5)	0.43	2 (0, 7)	3 (1, 5)	0.31
30 h	2 (0, 5)	2 (0, 10)	0.40	2 (0, 5)	2 (1, 10)	0.32
36 h	2 (0 6)	2 (0, 5)	0.86	2 (0, 6)	2 (1, 5)	0.69
42 h	2 (0, 5)	2.5 (0, 5)	0.19	2 (0, 5)	3 (0, 5)	0.24
48 h	2 (0, 5)	2 (0, 5)	0.72	2 (0, 5)	2 (0, 5)	0.47

All values were shown as median and range.

**Table 5 biomedicines-13-01784-t005:** Comparison of NRS pain scores after laparoendoscopic single-site surgery in patients exposed to CO_2_ for less than one hour.

Postoperative Time	Intention-to-Treat			Per Protocol		
	Celecoxib (n = 19)	Placebo (n = 15)	*p* Value	Celecoxib (n = 17)	Placebo (n = 13)	*p* Value
0 h	5 (1, 8)	5 (3, 7)	0.82	3 (1, 8)	5 (3, 7)	0.81
6 h	3 (1, 8)	3 (0, 5)	0.71	3 (1, 5)	3 (1, 5)	0.39
12 h	2 (1, 6)	3 (1, 5)	0.34	2 (1, 6)	3 (1, 5)	0.33
18 h	2 (1, 4)	2 (0, 4)	0.55	2 (1, 4)	2 (0, 4)	0.47
24 h	3 (0, 7)	2 (0, 5)	0.46	3 (0, 7)	2 (1, 5)	0.64
30 h	2 (0, 5)	2 (0, 5)	0.52	2 (0, 5)	1 (1, 5)	0.70
36 h	2 (1, 6)	1 (0, 4)	0.27	2 (1, 6)	1 (1, 4)	0.38
42 h	2 (0, 5)	2 (0, 3)	0.47	2 (0, 5)	2 (0, 3)	0.35
48 h	2 (0, 5)	2 (0, 3)	0.44	2 (0, 5)	2 (0, 3)	0.50

All values were shown as median and range.

**Table 6 biomedicines-13-01784-t006:** Comparison of NRS pain scores after laparoendoscopic single-site surgery in patients exposed to CO_2_ for more than one hour.

Postoperative Time	Intention-to-Treat			Per Protocol		
	Celecoxib (n = 11)	Placebo (n = 17)	*p* Value	Celecoxib (n = 10)	Placebo (n = 12)	*p* Value
0 h	4 (2, 9)	6 (3, 10)	0.36	4 (2, 0)	6 (3, 10)	0.13
6 h	2 (0, 4)	3 (1, 5)	0.09	2 (1, 4)	4 (2, 5)	0.02
12 h	2 (1, 5)	3 (1, 5)	0.15	1.5 (1. 4)	3 (1, 5)	0.03
18 h	1 (1, 5)	2 (1, 6)	0.02	1 (1, 4)	2 (1, 6)	0.01
24 h	2 (1, 4)	3 (1, 5)	0.05	2 (1, 3)	3 (2, 5)	0.02
30 h	2 (0, 3)	3 (1, 10)	0.08	2 (0, 3)	3 (1, 10)	0.04
36 h	1.5 (0, 3)	2.5 (1, 5)	0.11	1.5 (0, 3)	2.5 (1, 5)	0.09
42 h	1 (0, 2)	3 (0, 5)	<0.01	1 (0, 2)	3 (1, 5)	<0.01
48 h	2 (1, 3)	2 (0, 5)	0.18	2 (1, 3)	2.5 (1, 5)	0.04

All values were shown as median and range.

**Table 7 biomedicines-13-01784-t007:** Postoperative outcomes.

Outcomes	Celecoxib (n = 30)	Placebo (n = 32)	*p* Value
Operation time (min, mean, SD)	71.9 ± 56.6	80.7 ± 51.1	0.52
Estimated blood loss (mL, mean, SD)	265.1 ± 226.1	198.6 ± 161.3	0.19
No. of transfusion (median, range)	0 (0, 1)	0 (0, 2)	0.29
No. of rescue analgesics (median, range)	0 (0, 4)	0 (0, 3)	0.62
Discontinuation of PCA use (n, %)	7 (23.3)	6 (18.8)	0.75
Duration of hospitalization (d, range)	5 (4, 9)	5 (4, 5)	0.86

Abbreviation: PCA, patient-controlled analgesia; SD, standard deviation.

## Data Availability

The data that support the findings of this study are available from the corresponding authors upon reasonable request. The data are not publicly available due to privacy or ethical restrictions.

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
