# Peer review of "Upregulation of 15-Hydroxyprostaglandin Dehydrogenase by Celecoxib to Reduce Pain After Laparoendoscopic Single-Site Surgery (POPCORN Trial): A Randomized Controlled Trial"

_biomedicines, 2025, doi:10.3390/biomedicines13071784_

Round 1
Reviewer 1 Report
Comments and Suggestions for Authors
Dear Authors, your randomized double-blind study, comparing the effect of celecoxib with placebo on postoperative pain, is interesting and includes a novel analysis of HPGD expression. You included a proper discussion, considering the strengths and the weak points of the study. I have a few comments that, in my opinion, could improve the paper:
- A short paragraph on the prostanoid role in pain would be appropriate in the introduction (it is present in the discussion but I would consider moving it to introduction as it explains the reasons for the study)
-Please include an explanation with appropriate references, why a dose of 400mg was administered and not, for example, 200mg?
- There is no information on the type of anesthesia during the procedures. Please add data on whether all patients were anesthetized in the same way. Were there any other analgesic drugs administered perioperatively? This could have an impact on the NRS results.
Author Response
Dear Authors, your randomized double-blind study, comparing the effect of celecoxib with placebo on postoperative pain, is interesting and includes a novel analysis of HPGD expression. You included a proper discussion, considering the strengths and the weak points of the study. I have a few comments that, in my opinion, could improve the paper:
Comment 1: A short paragraph on the prostanoid role in pain would be appropriate in the introduction (it is present in the discussion but I would consider moving it to introduction as it explains the reasons for the study)
Response 1: Good recommendation! As you recommended, we moved the short paragraph to the Introduction (Line 64-69).
Comment 2: Please include an explanation with appropriate references, why a dose of 400mg was administered and not, for example, 200mg?
Response 2: Thank you for your kind comment. We inserted the reason as follows in “2.5. Surgical procedures in 2. Materials and Methods”
: In this study, we used 400 mg of celecoxib to evaluate the role of COX-2 inhibitor on postoperative pain because some studies had shown that 400 mg of celecoxib might be more effective than its baseline dose (200 mg) [24, 25].
Comment 3: There is no information on the type of anesthesia during the procedures. Please add data on whether all patients were anesthetized in the same way. Were there any other analgesic drugs administered perioperatively? This could have an impact on the NRS results.
Response 3: We appreciated your recommendation. Before surgery, placebo or celecoxib (400 mg) were applied based on randomization, and postoperatively, PCAs were applied to all patients, and removed if PONV occurred. We added it as follows in “2.5. Surgical procedures in 2. Materials and Methods”
: Postoperatively, PCAs were applied to all patients, and removed if PONV occurred.

Reviewer 2 Report
Comments and Suggestions for Authors
Thank you for the opportunity to read the manuscript titled: Upregulation of 15-Hydroxyprostaglandin Dehydrogenase by Celecoxib to Reduce Pain After Laparoendoscopic Single-site Surgery (POPCORN Trial): A Randomized Controlled Trial” by Kyung Hee Han and co-authors.
This manuscript reports on a randomized, double-blind, placebo-controlled pilot study investigating whether preoperative administration of celecoxib (400 mg) reduces postoperative pain after laparoendoscopic single-site (LESS) surgery for benign gynecologic conditions, via upregulation of 15-hydroxyprostaglandin dehydrogenase (HPGD) in peritoneal tissue. The authors evaluate gene expression changes (via qRT-PCR, IHC, and ISH), correlate them with COâ‚‚ exposure duration, and analyze their association with numeric rating scale (NRS) pain scores.
The findings suggest that celecoxib increases HPGD mRNA expression in patients exposed to COâ‚‚ for more than 60 minutes and results in lower NRS pain scores in this subgroup.
The study presents a novel hypothesis linking COX-2 inhibition to HPGD upregulation as a mediator of analgesia, rather than the classical pathway of prostaglandin synthesis inhibition alone. The work aligns well with efforts to personalize perioperative pain management and limit opioid use.
The study addresses a clinically meaningful question and presents interesting preliminary findings. However, the mechanistic claims are not yet fully supported by the data, and the subgroup-based findings should be more cautiously interpreted. Addressing these issues through clarification, additional discussion, and more precise language would significantly enhance the manuscript's scientific rigor and impact.
Major Concerns
- While mRNA expression of HPGD increased in the celecoxib group exposed to COâ‚‚ for >60 minutes, IHC results were negative for protein expression, and ISH showed variable expression changes. The conclusion that celecoxib “activates” HPGD is potentially overstated given the absence of protein-level confirmation and unclear functional activity. The authors should clarify whether increased mRNA expression was accompanied by functional enzyme activity, discuss potential post-transcriptional regulation or degradation of HPGD protein, and use more precise language such as “upregulates mRNA expression of HPGD” unless protein function is verified.
- The primary finding (celecoxib reduces pain and increases HPGD expression) is driven by a small subgroup (n=12 celecoxib patients with COâ‚‚ >60 min). This raises concerns about statistical power and type I error. Furthermore, no corrections for multiple testing are described across repeated measures and gene panels.
- The subgroup analysis should be interpreted cautiously and ideally reported as exploratory. This should be emphasized in the discussion.
- The rationale for choosing the five candidate genes is only briefly described. The initial gene screening (in n=5 celecoxib vs n=5 placebo patients) is too underpowered to justify generalization. It would also be useful to discuss why no differences were observed for PTGES/PTGIS/AKR1C3 and whether this challenges the overall hypothesis.
- COâ‚‚ exposure stratification: Was this a pre-specified subgroup analysis? If not, please state it clearly and discuss risk of bias.
- Please clarify whether the study was conducted as double-blind, or if blinding was limited to participants only.
Minor Comments
Figures:
Figures 3–4 could benefit from clearer legends and group identifiers.
Author Response
Thank you for the opportunity to read the manuscript titled: Upregulation of 15-Hydroxyprostaglandin Dehydrogenase by Celecoxib to Reduce Pain After Laparoendoscopic Single-site Surgery (POPCORN Trial): A Randomized Controlled Trial” by Kyung Hee Han and co-authors.
This manuscript reports on a randomized, double-blind, placebo-controlled pilot study investigating whether preoperative administration of celecoxib (400 mg) reduces postoperative pain after laparoendoscopic single-site (LESS) surgery for benign gynecologic conditions, via upregulation of 15-hydroxyprostaglandin dehydrogenase (HPGD) in peritoneal tissue. The authors evaluate gene expression changes (via qRT-PCR, IHC, and ISH), correlate them with COâ‚‚ exposure duration, and analyze their association with numeric rating scale (NRS) pain scores.
The findings suggest that celecoxib increases HPGD mRNA expression in patients exposed to COâ‚‚ for more than 60 minutes and results in lower NRS pain scores in this subgroup.
The study presents a novel hypothesis linking COX-2 inhibition to HPGD upregulation as a mediator of analgesia, rather than the classical pathway of prostaglandin synthesis inhibition alone. The work aligns well with efforts to personalize perioperative pain management and limit opioid use.
The study addresses a clinically meaningful question and presents interesting preliminary findings. However, the mechanistic claims are not yet fully supported by the data, and the subgroup-based findings should be more cautiously interpreted. Addressing these issues through clarification, additional discussion, and more precise language would significantly enhance the manuscript's scientific rigor and impact.
Major Concerns
Comment 1. While mRNA expression of HPGD increased in the celecoxib group exposed to COâ‚‚ for >60 minutes, IHC results were negative for protein expression, and ISH showed variable expression changes. The conclusion that celecoxib “activates” HPGD is potentially overstated given the absence of protein-level confirmation and unclear functional activity. The authors should clarify whether increased mRNA expression was accompanied by functional enzyme activity, discuss potential post-transcriptional regulation or degradation of HPGD protein, and use more precise language such as “upregulates mRNA expression of HPGD” unless protein function is verified.
Response 1. Thank you for your kind comment. We added your comment in “4. Discussion” as follows: “Fourth, we can only confirm that HPGD gene expression may be upregulated with in-creasing CO2 exposure time in the peritoneum, but cannot conclude whether this results in direct activation of HPGD gene via increased CO2 exposure time because we did not con-duct the measurement of post-transcriptional regulation of HPGD by increased mRNA expression” and the term “activate” has been changed into “upregulate” in this manuscript.
Comment 2. The primary finding (celecoxib reduces pain and increases HPGD expression) is driven by a small subgroup (n=12 celecoxib patients with COâ‚‚ >60 min). This raises concerns about statistical power and type I error. Furthermore, no corrections for multiple testing are described across repeated measures and gene panels.
Response 2. Good point! We added the limitation in “4. Discussion” as follows: “First, POPCORN trial was a pilot study due to the lack of sample size determination. Although the level 2 increase in HPGD expression aligned with expectations between the celecoxib and placebo groups, this comparison was based on only 28 patients exposed to CO2 for more than 60 minutes, which is less than half of the calculated sample size. Since there was no validation by repeated assessments, the results of POPCONRN trial are limited by low statistical power and the possibility of false positives.”
Comment 3. The subgroup analysis should be interpreted cautiously and ideally reported as exploratory. This should be emphasized in the discussion.
Response 3. Your comment is very important. So, we mentioned it in “4. Discussion” as follows: “Second, the results of this study may be affected by variations in the average duration of CO2 exposure due to variations in the number of patients. Thus, it is necessary to validate the results in a clinical trial with a larger number of patients.”
Comment 4. The rationale for choosing the five candidate genes is only briefly described. The initial gene screening (in n=5 celecoxib vs n=5 placebo patients) is too underpowered to justify generalization. It would also be useful to discuss why no differences were observed for PTGES/PTGIS/AKR1C3 and whether this challenges the overall hypothesis.
Response 4. Good point! The POPCORN trial was a pilot study and was conducted in the smallest number of patients expected to be clinically valid due to a lack of relevant studies for calculating the sample size. Thus, we mentioned the limitations of this study, including the lack of power of initial screening to identify candidate genes in a limited number of patients, and the inability to conclude that some genes including PTGES, PTGIS, AKR1C3 are not necessarily induced by CO2 exposure.
“Third, the initial screening to identify candidate genes related to CO2 exposure was un-derpowered due to a limited number of patients in this pilot study, and we could not con-clud that some genes including PTGES, PTGIS and AKR1C3 are not necessarily induced by CO2 exposure.”
Comment 5. COâ‚‚ exposure stratification: Was this a pre-specified subgroup analysis? If not, please state it clearly and discuss risk of bias.
Response 5. Good question! We conducted subgroup analyses on a CO2 exposure time of 60 minutes because an average peritoneal tissue exposure time to CO2 was 60.4 minutes mentioned in “3.1 Clinicopathologic characteristics in 3. Results”. Since the results of this study may be affected by variations in the average duration of CO2 exposure due to variations in the number of patients, it is necessary to validate the results in a clinical trial with a larger number of patients. We added it in “4. Discussion” as follows: “Second, the results of this study may be affected by variations in the average duration of CO2 exposure due to variations in the number of patients. Thus, it is necessary to validate the results in a clinical trial with a larger number of patients.”
Comment 6. Please clarify whether the study was conducted as double-blind, or if blinding was limited to participants only.
Response 6. Good question! We already mentioned that This study was designed as a randomized, double-blind, placebo-controlled pilot study in “2.1. Study design in 2. Materials and Methods”
Minor Comments
Comment 7. Figures: Figures 3–4 could benefit from clearer legends and group identifiers.
Response 7. Thank you for your comment. We revised them more clearly.

Round 2
Reviewer 1 Report
Comments and Suggestions for Authors
The Authors applied necessary adjustments to the paper except for the last remark. Because the study is on postoperative pain, I strongly recommend adding full information on the intraoperative anesthesia and any analgesic drugs that were administered, which the authors still did not include but which could have an impact on the study results (the information on postoperative analgesia was added but not on intraoperative treatment) - probably all patients had general anesthesia, which should be mentioned in the Materials and Methods section, as well as information on what analgesic drugs were administered during the surgery - only opioids? Other non-steroidal anti-inflammatory drugs?
Author Response
Comment 1.
The Authors applied necessary adjustments to the paper except for the last remark. Because the study is on postoperative pain, I strongly recommend adding full information on the intraoperative anesthesia and any analgesic drugs that were administered, which the authors still did not include but which could have an impact on the study results (the information on postoperative analgesia was added but not on intraoperative treatment) - probably all patients had general anesthesia, which should be mentioned in the Materials and Methods section, as well as information on what analgesic drugs were administered during the surgery - only opioids? Other non-steroidal anti-inflammatory drugs?
Response 1
Thank you for your kind question. In this study, intraoperative pain was controlled by fentanyl, which was also used in PCA for controlling postoperative pain. We mentioned them as follows in 2.5. Surgical procedures
“During the induction of general anesthesia, an initial intravenous dose of fentanyl ranging from 50 to 100 micrograms was administered. Throughout the surgery, additional doses of fentanyl were given at intervals of 30to 60 minutes, depending on the patient’s needs and the duration of procedure. After the surgery, once the patients regained consciousness, PCA was implemented to manage postoperative pain in all patients. For PCA administration, 100 micrograms of fentanyl was diluted in 2 ml of normal saline and administered intravenously. Additionally, a continuous infusion was prepared by diluting 1000 micrograms of fentanyl in 80 ml of normal saline, which was delivered at a constant rate. To allow the patient to manage breakthrough pain, the PCA device was programed so that the patient could self-administer a 50-microgam bolus dose of fentanyl when needed. This approach provided both continuous baseline pain control and the flexibility for the patient to address acute pain episodes effectively. Moreover, PICA was removed if PONV occurred.”

Reviewer 2 Report
Comments and Suggestions for Authors
The authors have made a genuine effort to comply with the reviewers’ previous requests and have clarified key points in the revised manuscript. While certain limitations remain, particularly regarding the mechanistic link between celecoxib and HPGD activation, the study’s pilot nature is now better acknowledged. I believe the manuscript is suitable for publication following minor editorial adjustments.
I recommend a small revision to the conclusion of the abstract to better reflect the preliminary nature of the findings. Suggested version:
“In this pilot study, celecoxib appeared to reduce postoperative pain and was associated with increased HPGD mRNA expression in peritoneal tissue of patients with prolonged COâ‚‚ exposure during LESS surgery. These exploratory findings warrant confirmation in larger trials with functional validation of HPGD expression (ClinicalTrials.gov, NCT03391570).”
Author Response
Comment 1
The authors have made a genuine effort to comply with the reviewers’ previous requests and have clarified key points in the revised manuscript. While certain limitations remain, particularly regarding the mechanistic link between celecoxib and HPGD activation, the study’s pilot nature is now better acknowledged. I believe the manuscript is suitable for publication following minor editorial adjustments.
I recommend a small revision to the conclusion of the abstract to better reflect the preliminary nature of the findings. Suggested version:
“In this pilot study, celecoxib appeared to reduce postoperative pain and was associated with increased HPGD mRNA expression in peritoneal tissue of patients with prolonged COâ‚‚ exposure during LESS surgery. These exploratory findings warrant confirmation in larger trials with functional validation of HPGD expression (ClinicalTrials.gov, NCT03391570).”
Response 1
Thank you for your valuable comment. We revised it as you recommended.
